# Substance use, psychiatric symptoms, personal mastery, and social support among COVID-19 long haulers: A compensatory model

Cheuk Chi Tam[1,2]*, Shan Qiao[1,2], Camryn Garrett[1,2], Ran Zhang[1,2], Atefeh Aghaei[1,2], Abhishek Aggarwal[1,2], Alain H. Litwin[3,4], Xiaoming Li[1,2]

1 South Carolina SmartState Center for Healthcare Quality, Arnold School of Public Health, University of South Carolina, Columbia, SC, United States of America, 2 Department of Health Promotion, Education and Behavior, Arnold School of Public Health, University of South Carolina, Columbia, SC, United States of America, 3 School of Health Research, Clemson University, Greenville, SC, United States of America, 4 Department of Medicine, School of Medicine Greenville, University of South Carolina, Greenville, SC, United States of America

* ctam@mailbox.sc.edu

**Data Availability Statement:** Data used in the current are in the dataset available from the Figshare database at https://figshare.com/articles/

## Abstract

### Background

Substance use has become a critical health concern during the COVID-19 pandemic, and emerging attention has been paid to people with the persistent symptoms of COVID-19 (COVID-19 long haulers) due to their high vulnerability. However, scant research has investigated their substance use and relevant psychosocial factors. The current study was to (1) examine substance use behaviors (i.e., legal drug use, illicit drug use, and non-medical use of prescription drugs); and (2) assessed their associations with psychiatric symptoms (i.e., depression, anxiety, and post-traumatic stress disorder) and psychosocial factors (i.e., personal mastery and social support) among COVID-19 long haulers.

### Methods

In January–March 2022, 460 COVID-19 long haulers (50% female), with an average age of 32, completed online surveys regarding their demographics, substance use, psychiatric symptoms, and psychosocial factors.

### Results

In the past three months, the most commonly used or non-medically used substances were tobacco (82%) for legal drugs, cocaine (53%) for illicit drugs, and prescription opioids (67%) for prescription drugs. Structural equation modeling suggested that psychiatric symptoms were positively associated with substance use behaviors ($\beta$s = 0.38 to .68, $p$s < 0.001), while psychosocial factors were negatively associated with substance use behaviors ($\beta$s = -0.61 to -0.43, $p$s < 0.001).

dataset/SPSS_dataset_of_substance_use_in_
COVID_long_haulers_sav/22285330.

**Funding:** The research was supported in part by
the by the National Institute of Allergy and
Infectious Diseases of the National Institutes of
Health under award number R01AI127203-5S1.
The content is solely the responsibility of the
authors and does not necessarily represent the
official views of the National Institutes of Health.
There was no additional external funding received
for this study.

**Competing interests:** The authors have declared
that no competing interests exist.

## Conclusion

Substance use is common in COVID-19 long haulers and psychiatric symptoms are the risk
factors. Personal mastery and social support appear to offer protection offsetting the psychi-
atric influences. Substance use prevention and mental health services for COVID-19 long
haulers should attend to personal mastery and social support.

## Introduction

The COVID-19 pandemic began in March 2020 and continues to persist two and a half years
later with the development of multiple virus variants. SARS-CoV-2 is a coronavirus virus that
in many causes respiratory symptoms (e.g., fever/chills) but also musculoskeletal (e.g., fatigue)
and gastrointestinal symptoms [1]. COVID-19 long-haulers are defined as those who experi-
ence the persistence of COVID-19 induced symptoms (Long COVID) for more than four
weeks after infection [2, 3]. As of July 6, 2022, it has been estimated 87,899,721 cumulative
COVID-19 cases worldwide [4]. Of those infected with COVID-19, 13.3% experience persis-
tent symptoms one month or longer after infection with 2.5% reporting symptoms persisting
for three or more months [5].

Due to the increasing burden of Long-COVID, attention has focused on the health issues of
COVID-19 long haulers. One emerging health concern is their vulnerability to substance use.
The COVID-19 literature has highlighted the exponential rates of individuals engaging in sub-
stance use (i.e., 13% increase since COVID-19) and drug overdose fatalities (i.e., 30% increase
from 2019) in the United States (US) [6, 7]. These increases have been found in the use of
numerous classes of drugs, including tobacco/cigarette products [8], alcohol [9], illicit drugs
[10–13], and non-medical use of prescription drugs [14]. Notably, the substance use issue
could be significant among COVID-19 long haulers for two reasons. First, those with history
of substance use disorder (SUD) are more likely to be infected by COVID-19 and develop
severe symptoms than those without a history of SUD [15, 16]. Second, the management of
persistent symptoms (e.g., headache and myalgias) requires many COVID-19 long haulers to
utilize prescription medications, which may interact with or develop a history of SUD. Existing
evidence has shown that COVID-19 long haulers are prescribed opioids and benzodiazepines
at higher rates (i.e., an extra 8–22 times per 1,000 patients) than patients without a COVID-19
diagnosis [17]. As such, the substance use literature turns its attention to COVID-19 long haul-
ers, and, indeed, a recent cohort study of long haulers found a high rate of problematic alcohol
use (45.5%) 2–4 months after their initial clinic visit, and such a rate even increased (71.8%) at
the 7–14 months follow-up [18]. Despite such a high vulnerability, available literature on vari-
ous substance uses in COVID-19 long haulers is still lacking. In efforts to address this gap, a
study that investigates substance use and its related factors among COVID-19 long haulers
appears to be warranted.

The compensatory model of psychological resilience [19] may offer a conceptual framework
to examine factors associated with substance use in COVID-19 long haulers. This model
depicts a psychosocial mechanism on maladaptive responses (e.g., substance use) to adverse
conditions (e.g., long COVID), suggesting that maladaptive responses occur under the influ-
ences of risk factors (e.g., emotional distress), while protective factors (e.g., psychosocial
resources) counteract to offset risk factors. This model has been widely applied within health
science research for understanding psychosocial influences on substance use (e.g., alcohol, cig-
arette, and illicit drugs) [20, 21]. Accordingly, given the tendency for reliance on maladaptive

responses, the understanding of substance use in COVID-19 long haulers would benefit from examining a compensatory model on a set of psychosocial risk and protective factors.

Psychiatric symptoms could be a robust risk factor for substance use among COVID-19 long haulers. The COVID-19 pandemic has taken a significant toll on mental health (e.g., persistently elevated rates of reported serious psychiatric symptoms since the onset of the pandemic; 12–19%) [22]. Due to this toll many individuals have turned to substance use to cope with their psychiatric distress [23]. The global literature has demonstrated that psychiatric symptoms (i.e., depression, anxiety, and post-traumatic stress disorder [PTSD]) were common triggers of use of substances (i.e., cigarettes, alcohol, and illicit drugs) during the COVID-19 pandemic [24–27]. Due to their substantially high risk for psychiatric symptoms, substance use coping among COVID-19 long haulers is concerning. A review and meta-analysis on post-COVID-19 studies documented the high prevalence of anxiety (7% to 63%), depression (4% to 31%), and PTSD (12% to 46.9%) among COVID-19 long haulers where the prevalence significantly increased over time (i.e., at the one-year follow-up) [28, 29]. Existing evidence also indicates that, compared to those who were not diagnosed with COVID-19, COVID-19 survivors reported significantly higher levels of depression and anxiety [30]. Despite these preliminary findings, scant research has examined the association between psychiatric symptoms and substance use among COVID-19 long haulers.

Personal mastery and social support could be core intra- and interpersonal resilient resources for COVID-19 long haulers, serving as protective factors against substance use. Personal mastery refers to individuals' perceived ability to have control over and adapt to the influential forces in their lives [31]. Personal mastery has been identified as an essential proxy of personal psychological strengths (e.g., self-esteem, sense of control, and positive appraisal of stress) and promoting resistance to substance use [32–35]. On the other hand, social support refers to perceived external possessions and resources (i.e., material, emotional, or tangible) that are assessable to individuals, within their social networks, to deal with stressors [36]. Social support has been associated with effective coping with distress and promoting resistance to substance use [37, 38]. Due to their promising effects, these two factors are viewed as particularly beneficial for people with chronic symptoms (e.g., COVID-19 long haulers). Literature on individuals with chronic illnesses (e.g., diabetes, lung disease, and rheumatoid arthritis) have indicated that high personal mastery and social support are associated with better management of stress related to persistent symptoms (e.g., pain, fatigue, or glycemic control) [39–41]. Within the COVID-19 literature, the protective roles of personal mastery and social support have been documented regarding the use of alcohol and tobacco products [42]. Given these merits in coping among those with other chronic conditions, personal mastery and social support could serve as protective factors among COVID-19 long haulers, further necessitating a study to examine their associations with substance use behaviors.

## The present study

Given the scarcity of studies investigating substance use and its related factors among COVID-19 long haulers, the present study aimed to (1) examine recent (i.e., past-three-month) use of substances across a variety of classes, including legal drug use (e.g., alcohol and tobacco products), illicit drugs (e.g., cocaine and inhalants), and non-medical use of prescription drugs (NMUPD; e.g., opioids and stimulants); (2) examine the associations between substance use behaviors and psychiatric/psychosocial factors among COVID-19 long haulers. Applying the compensatory model, it was hypothesized that (1) psychiatric symptoms, including depression, anxiety, and PTSD, act as risk factors with a positive association with substance use and (2) personal mastery and social support act as protective factors with a negative association with substance use (see Fig 1).

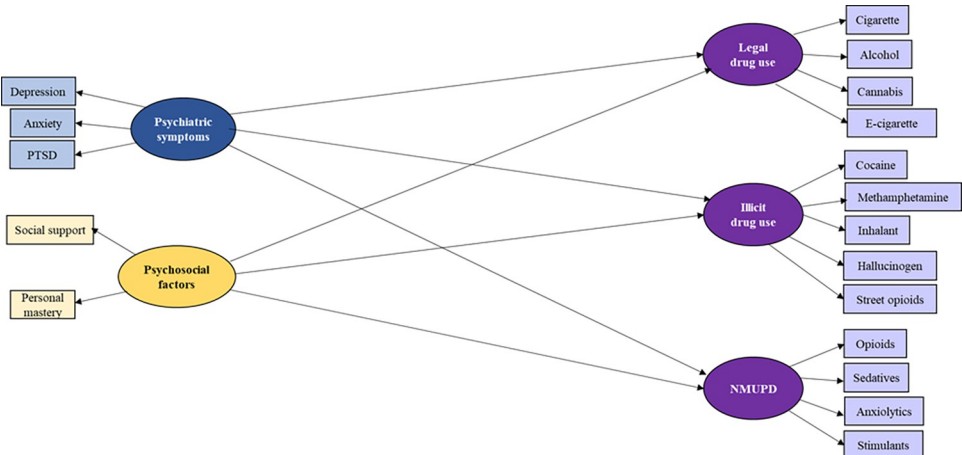

**Fig 1. Hypothesized model of the associations of psychiatric symptoms and psychosocial factors with substance use variables among COVID long haulers.**

## Methods

### Participants and procedures

Participants were recruited utilizing a convenience sampling approach, specifically, voluntary response and purposive sampling. Participants were able to participate in the study by voluntarily responding to calls within online support groups on social media (i.e., Facebook) topically related to COVID-19. The study was conducted over a two-month period from January 13, 2022 to March 26, 2022. The inclusion criteria included only those participants: (1) aged 18 years or older, (2) living in the United States, and (3) who have been infected with COVID-19 and experienced at least one persistent symptom four weeks or longer after the COVID-19 diagnosis. In efforts to purposively target those who were likely to meet the inclusion criteria, the study groups utilized to post calls for participants were identified through searches of online support groups highlighting COVID-19 related topics.

The study personnel contacted the group administrators for preemptive permission to post recruitment materials. Administrators were contacted a maximum of three times. If after three attempts to contact no response was given, then the posts were submitted to the group's page. Those pages with lesser regulation allowed the post to be made directly. Those pages with more regulations required administrator approval of the post before it was published to group members. Once the recruitment materials (i.e., brief study description, survey link, and flyer) were posted on the Facebook pages, potential participants were able to access the online 20-minute survey as distributed through the survey platform Redcap. Upon obtaining online written informed consent, the eligible participants were able to self-administer the survey and navigate through the online survey answering only those questions they wished to, voluntarily. Participants were able to withdraw at any point by exiting the survey window and only those surveys completed in full were included in the analyses. Upon completion participants were able to opt-in to a raffle-based incentive for a gift card (USD $40). No explicitly identifying information was collected and participant responses were assigned a participant number to maintain confidentiality.

The study protocol was reviewed and approved in accordance with 45 CFR 46.104(d)(2) and 45 CFR 46.111(a)(7) by the University of South Carolina Institutional Review Board and was deemed as exempt from Human Research Subject Regulations due to the minimal risks posed to participants. Data were deidentified and collected with the informed consent of participants prior to their participation in the study.

## Measures

**Demographics.** Demographic information collected included age, gender, race/ethnicity, education (e.g., a high school degree or equivalent), and annual income (USD).

**Substance use.** Participants were asked to respond on a 13-item measure to report substances that they used in the past three months at the time of survey response. This measure was adapted from the Alcohol, Smoking, and Substance Involvement Screening Test (ASSIST) [43]. The measure includes individual items for drugs or medications spanning a variety of classes, including legal drugs (e.g., tobacco), illicit drugs (e.g., cocaine), and prescription opioids (e.g., OxyContin), prescription sedatives (e.g., Ambien), prescription anxiolytics (e.g., Ativan), and prescription stimulants (e.g., Ritalin). Questions for legal and illicit drug items were about the frequency of use, while these for prescription drugs were about the frequency of use without a physician's prescription. Participants responded to each item on a 5-option Likert scale with response options ranging from 1 (never) to 5 (almost daily). For use in the descriptive statistics, responses were dichotomized to report "ever use" (0 = never; 1 = have used/non-medical used) and "weekly use or more" (0 = never, 1 = have used/non-medical used once a week or more often) on each individual drug item. Continuous scores on each item were used for data analyses.

**Anxiety.** Anxiety was assessed with the General Anxiety Disorder (GAD-7) 7-item survey which included 4 response options ranging from 0 (not at all) to 4 (nearly every day) [44]. Participants were asked to indicate to what extent they felt nervous or worried too much about a situation in the past two weeks (e.g., "*Not being able to stop or control worrying*"). The sum scores were generated, with higher scores indicating greater levels of anxiety. Cronbach's alpha for this scale was 0.59 in the current study.

**Depression.** Depression was assessed with the Patient Health Questionnaire 9-item (PHQ-9) scale including 4 response options ranging from not at all (0) to nearly every day (3) [45]. Items asked about the level of depressive moods that participants experienced in the past two weeks (e.g., "*Feeling down, depressed, and hopeless*"). Sum scores were calculated, with a higher score showing a higher level of depression. Cronbach's alpha for this scale was 0.57 in the current study.

**PTSD.** PTSD was assessed with an adapted version of the Primary Care PTSD Screen for the DSM-5 5-item scale (PC-PTSD-5) with binary response options yes or no [46]. Participants were asked to report how stressful life events during the COVID-19 pandemic affected them in the past month at the time of survey response (e.g., "*Had nightmares about event[s] or thought about the event[s] when you did not want to*"). Sum scores were generated, with higher scores representing greater levels of PTSD. Cronbach's alpha for this scale was 0.48 in the current study.

**Personal mastery.** Personal mastery was measured with a 4-item subscale of the Sense of Control Scale. Participants were asked to select one of 7 response options ranging from strongly disagree (1) to strongly agree (4) [47]. Items assessed individual sentiments of ability and confidence to complete general things as well as evaluated their certainty towards the future (e.g., "*What happens to me in the future mostly depends on me*"). Sum scores were calculated, with higher scores indicating greater levels of personal mastery. Cronbach's alpha for this scale was 0.54 in the current study.

**Social support.** Social support was assessed using the modified Medical Outcomes Study Social Support Survey (mMOS-SS) 8-item scale [48]. Items assessed the presence of support in participants' lives through their perceptions of having others around them to provide support in times of need (e.g., "*Someone to take you to the doctor if you need it*"), provide understanding (e.g., "*Someone who understand your problems*"), and to have a good time with (e.g.,

"*Someone to have a good time with*"). Response options to the items ranged from none of the time (1) to all of the time (5). Sum scores were generated, with higher scores indicating higher levels of present social support. Cronbach's alpha for this scale was 0.61 in the current study.

## Statistical analysis

Data were screened in terms of multivariate outliers, missing data, and normality. Descriptive statistics were employed on demographic factors, 13 individual items of substance use, psychiatric symptoms (i.e., depression, anxiety, and PTSD), personal mastery, and social support.

We used structural equation modeling (SEM) with the maximum likelihood estimation (ML) to examine the hypothesized associations. As guided by Anderson and Gerbing [49] (1988), we used confirmatory factor analysis (CFA) to examine a measurement model, which evaluated the latent structures of study variables (i.e., legal drug use, illicit drug use, NMUPD, psychiatric symptoms, and psychosocial factors). The model was anticipated to be trimmed in accordance with the modification indices (cutoff value > 10) of the suggested relationships if this was supported by previous research. The hypothesized associations of psychiatric symptoms and psychosocial factors with legal drug use, illicit drug use, and NMUPD were then examined using structural modeling when controlling for demographics (Fig 1). The goodness of model fit for the measurement model and structural model was determined according to various indices including the *Chi-square value*/*degree of freedom* (*df*), the *comparative fit index* (*CFI*), the *Tucker-Lewis index* (*TLI*), the *root mean square of approximation* (*RMSEA*), and the *standardized root mean square of residual* (*SRMR*). As suggested by Meyers, Gamst [50], a model has an adequate or good fit to data when *Chi-square value*/*df* is lower than 3 (good), *CFI* is higher than 0.90 (adequate) or 0.95 (good), *TLI* is higher than 0.90 (adequate) or 0.95 (good), *RMSEA* is lower than 0.08 (adequate) or 0.05 (good), and *SRM*R is lower than 0.08 (good).

## Results

### Demographics and descriptive statistics

The mean age of COVID-19 long haulers was 32 years (*SD* = 6.19). About a half of the sample were female (50.22%). The majority of participants were White/Caucasian (70.00%), followed by Black/African American (17.20%), and Hispanic/Latino (8.90%). The majority had a bachelor's degree (59.57%) and reported an annual income between $50,000 and $100,000 (56.09%). Participants reported a mean score of 10.18 (*SD* = 3.51) for anxiety, 13.00 (*SD* = 3.94) for depression, 2.51 (*SD* = 1.42) for PTSD, 17.38 (*SD* = 4.05) for personal mastery, and 25.28 (*SD* = 4.68) for social support (see Table 1).

### Substance use

In the past three months, the commonly used legal drugs were tobacco or cigarettes (81.74% ever use; 33.04% weekly use), followed by alcohol (80.87% ever use; 32.82% weekly use), e-cigarettes or vaping (69.57% ever use; 23.26% weekly use), and cannabis (35.87% ever use; 25.43% weekly use). Cocaine (52.39% ever use; 18.91% weekly use) was the most commonly used illicit drug, followed by inhalants (50.22% ever use; 19.34% weekly use), methamphetamine (47.17% ever use; 20% weekly use), hallucinogens (45.87% ever use; 17.61% weekly use), and street opioids (44.57% ever use; 15% weekly use). The commonly and non-medically used prescription drugs were opioids (67.39% ever use; 17.83% weekly use), followed by sedatives (65% ever us; 25% weekly use), anxiolytics (61.74% ever use; 20.65% weekly use), and stimulants (58.91% ever use; 21.52% weekly use).

**Table 1. Descriptive statistics of demographic characteristics, substance use, psychiatric symptoms, and psychosocial factors among COVID-19 long haulers ($n$ = 460).**

| Variables | Mean (SD) or n (%) |
|---|---|
| Age, Mean (SD) | 32 (6.19) |
| Gender, n (%) | |
| Female | 231 (50.22%) |
| Male | 225 (48.91%) |
| Other/ Prefer not to say | 4 (0.87%) |
| Race/ Ethnicity, n (%) | |
| White/Caucasian | 322 (70.00%) |
| Black/African American | 79 (17.20%) |
| Hispanic/Latino | 41(8.90%) |
| Asian | 4 (0.90%) |
| Native American/Alaskan Native; Native Hawaiian Pacific Islander | 18 (3.90%) |
| Education, n (%) | |
| Highschool degree or equivalent | 14 (3.04%) |
| Some college, but no degree | 94 (20.43%) |
| Associates degree | 60 (13.04%) |
| Bachelor's degree | 274 (59.57%) |
| Post graduate degree or above | 18 (3.91%) |
| Annual Income, n (%) | |
| < $10,000 to $24,999 | 27 (5.87%) |
| $25,000 to $49,999 | 164 (35.65%) |
| $50,000 to $100,000 | 258 (56.09%) |
| >$100,000 | 11 (2.39%) |
| **Substance use in the past three months** | |
| **Legal drug use** | |
| Tobacco/cigarettes, Mean (SD) | 2.87 (1.30) |
| Ever use, n (%) | 376 (81.74%) |
| Weekly use or more, n (%) | 152 (33.04%) |
| Alcohol, Mean (SD) | 2.86 (1.27) |
| Ever use, n (%) | 372 (80.87%) |
| Weekly use or more, n (%) | 151 (32.82%) |
| Cannabis, Mean (SD) | 2.4 (1.36) |
| Ever use, n (%) | 295 (35.87%) |
| Weekly use or more, n (%) | 117 (25.43%) |
| E-cigarettes/ vaping, Mean (SD) | 2.45 (1.29) |
| Ever use, n (%) | 320 (69.57%) |
| Weekly use or more, n (%) | 107 (23.26%) |
| **Illicit drug use** | |
| Cocaine, Mean (SD) | 2.12 (1.31) |
| Ever use, n (%) | 241 (52.39%) |
| Weekly use or more, n (%) | 87 (18.91%) |
| Methamphetamine, Mean (SD) | 2.07 (1.38) |
| Ever use, n (%) | 217 (47.17%) |
| Weekly use or more, n (%) | 92 (20%) |
| Inhalants, Mean (SD) | 2.1 (1.35) |
| Ever use, n (%) | 231 (50.22%) |
| Weekly use or more, n (%) | 89 (19.34%) |

*(Continued)*

**Table 1.** (Continued)

| Variables | Mean (SD) or n (%) |
|---|---|
| Hallucinogens, *Mean* (*SD*) | 2.02 (1.34) |
| Ever use, *n* (%) | 211 (45.87%) |
| Weekly use or more, *n* (%) | 81 (17.61%) |
| Street opioids, *Mean* (*SD*) | 1.93 (1.26) |
| Ever use, *n* (%) | 205 (44.57%) |
| Weekly use or more, *n* (%) | 69 (15%) |
| **NMUPD** | |
| Use Prescription opioids without prescription, *Mean* (*SD*) | 2.3 (1.22) |
| Ever use, *n* (%) | 310 (67.39%) |
| Weekly use or more, *n* (%) | 82 (17.83%) |
| Use Prescription sedatives without prescription, *Mean* (*SD*) | 2.4 (1.35) |
| Ever use, *n* (%) | 299 (65%) |
| Weekly use or more, *n* (%) | 115 (25%) |
| Use Prescription anxiolytics without prescription, *Mean* (*SD*) | 2.29 (1.31%) |
| Ever use, *n* (%) | 284 (61.74%) |
| Weekly use or more, *n* (%) | 95 (20.65%) |
| Use Prescription stimulants without a prescription, *Mean* (*SD*) | 2.31 (1.37) |
| Ever use, *n* (%) | 271 (58.91%) |
| Weekly use or more, *n* (%) | 99 (21.52%) |
| **Psychiatric symptoms** | 10.18 (3.51) |
| Anxiety, *Mean* (*SD*) | 13 (3.94) |
| Depression, *Mean* (*SD*) | |
| PTSD, *Mean* (*SD*) | 2.51 (1.42) |
| **Psychosocial factors** | |
| Personal mastery, *Mean (SD)* | 17.38 (4.05) |
| Social support, *Mean (SD)* | 25.28 (4.68) |

Note: *SD = Standard Deviation.*

## Measurement model

CFA was used to examine the measurement model among latent structures for study variables. The initial model suggested a generally good fit to data (*CFI* = 0.96; *TLI* = 0.95; *RMSEA* = 0.04; *SRMR* = 0.06). However, in accordance with the modification indices, the model could be improved by adding a correlation of manifest factors between alcohol and cigarette use. This correlation is in line with COVID-19 research, which indicates that alcohol consumption was higher in smokers during the pandemic [51]. We reran CFA on the new measurement model as suggested.

As shown in Fig 2, the results suggest that all manifest factor variables were significantly loaded onto their corresponding latent factors (*p*s < 0.001). The latent factors for three classes of substance use were positively inter-correlated (*r*s = 0.84–0.96; *p*s < 0.001). Psychiatric symptoms were positively correlated with legal drug use (*r* = 0.44, *p* < 0.001), illicit drug use (*r* = 0.16, *p* = 0.004), and NMUPD (*r* = 0.48, *p* < 0.001). Psychosocial factors were negatively associated with legal drug use (*r* = -0.18, *p* < 0.001), illicit drug use (*r* = -0.44, *p* = 0.04), and NMUPD (*r* = -0.20, *p* = 0.003). Psychiatric symptoms were positively correlated with psychosocial factors (*r* = 0.49, *p* < 0.001). The model-fit indices were 0.97 for *CFI*, 0.96 for *TLI*, 0.03 for *RMSEA*, and 0.06 for *SRMR*, suggesting an overall good fit to the data.

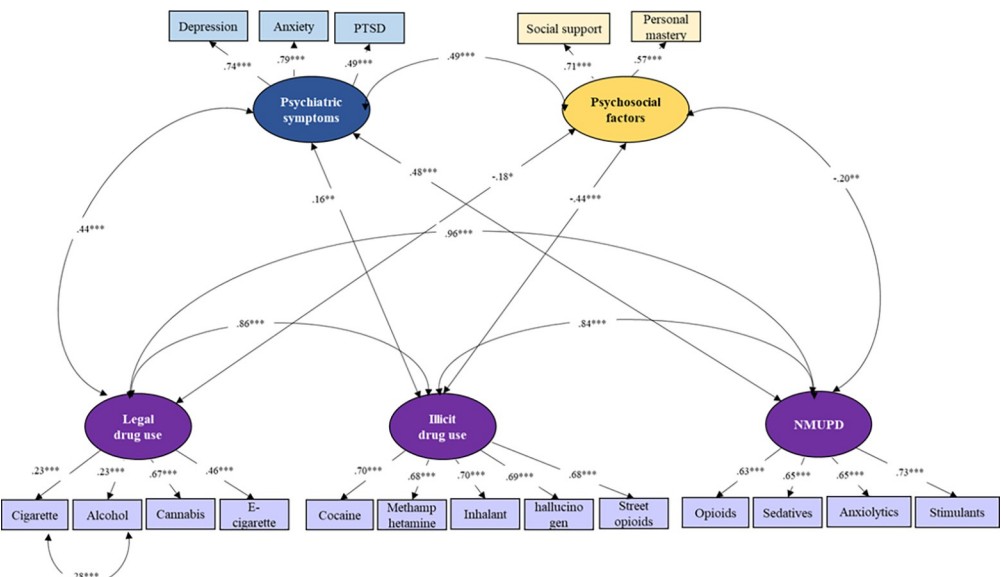

**Fig 2. Measurement model among substance use variables, psychiatric symptoms, and psychosocial factors among COVID long haulers (*n* = 460).**

## Structural equation modeling

SEM was used to examine the hypothesized model among substance use psychiatric symptoms, and psychosocial factors, when controlling for demographic factors (i.e., age, gender, race/ethnicity, and annual income). The standardized regression coefficients between latent variables are presented in Fig 3. The latent structures in the measurement model were applied to the structural model. Results suggest that the structural model explained 69% variance in legal drug use, 57% in illicit drug, and 74% in NMUPD. The model-fit indices suggest an overall good fit to the data (*CFI* = 0.93, *TLI* = 0.91, *RMSEA* = 0.05, *SRMR* = 0.07, *Chi-square/df* = 1.96). The regression coefficients suggest that psychiatric symptoms were positively associated with legal drug use ($\beta$ = 0.59, *p* < 0.001), illicit drug use ($\beta$ = 0.38, *p* < 0.001), and

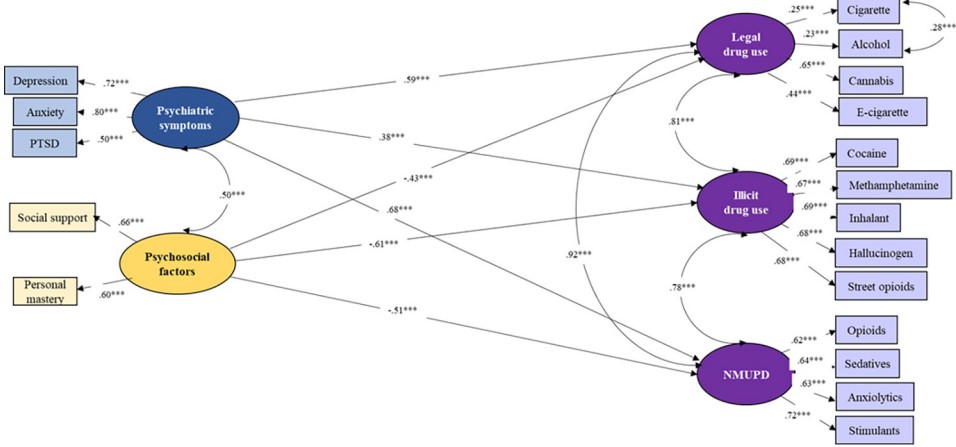

**Fig 3. Structural equation modeling on the associations of psychiatric symptoms and psychosocial factors with substance use variables among COVID long haulers, after controlling for demographics (i.e., age, gender, race/ethnicity, and annual income) (*n* = 460).**

NMUPD ($\beta$ = 0.68, $p$ < 0.001), indicating psychiatric symptoms (i.e., depression, anxiety, and PTSD) as risk factors for substance use among COVID-19 long haulers. In contrast, psychosocial factors were negatively associated with legal drug use ($\beta$ = -0.43, $p$ < 0.001), illicit drug use ($\beta$ = -0.51, $p$ < 0.001), and NMUPD ($\beta$ = -0.61, $p$ < 0.001), suggesting that personal mastery and social support were protective factors for substance use.

## Discussion

The present study explored the patterns of using and misusing substances across different classes (i.e., legal, illicit, and prescription drugs) among COVID-19 long haulers and examined psychosocial risk (i.e., depression, anxiety, and PTSD) and protective factors (i.e., social support and personal mastery) associated with their substance use behaviors. Although emerging attention has been paid to psycho-behavioral health in COVID-19 long haulers, to the best of our knowledge, this is one of the first attempt to document substance use profiles in this population and investigate the relevant psychosocial determinants.

Our findings suggest overall high rates of substance use in COVID-19 long haulers, with more than half of participants reporting using or misusing any classes of substances in the past three months and more than 15% reporting heavy use (i.e., weekly use or more). This reflects the previous findings of high vulnerability, revealing that people infected with COVID-19 were 20%-34% more likely to develop substance use disorders (i.e., opioid use disorder and alcohol use disorder) compared to those without infection [52]. Notably, our results seem to be higher than the rates in COVID-19 survivors (e.g., 15% current smoking; 2.6% to 23% past-year alcohol and drug use disorders) [52, 53] or people living with chronic conditions (13% substance use disorder) [54]. Specifically, we found that the commonly used substances were tobacco or alcohol products (i.e., 70% - 82%, cigarettes/e-cigarettes; 81%, alcohol). This is aligned with recent clinical research, indicating that 72% of COVID-19 long haulers reporting past-year problematic alcohol use [18]. It is noteworthy that our data also show high prevalence in NMUPD (e.g., prescription opioids; 68% ever use, 18% weekly use or more) and illicit drug use (e.g., cocaine; 52% ever use, 33% weekly use or more). This merits a particular concern given its negative influences on their persistent symptoms (e.g., respiratory infections) [55–57]. These findings suggest that COVID-19 long haulers appear to be a subgroup in people infected by COVID-19 who are significantly susceptible to substance use, implying the necessity to include substance use assessment in risk screen tools for patients with persistent symptoms of COVID-19 at the clinical settings [15, 16].

Our findings suggest that psychiatric distress (i.e., depression, anxiety, and PTSD) is a risk factor significantly associated with a variety of substance use behaviors, including legal drug use, illicit drug use, and NMUPD, in COVID-19 long haulers. This is an extension to previous research, which showed that individuals turned to substance use to manage psychological distress associated with the COVID-19 pandemic or chronic conditions (e.g., HIV, Hepatitis C, and chronic pain) [24, 58–60]. This may be explained by the self-medication theory of addiction [61]. The theory posits that individuals with conditions (e.g., persistent symptoms) that limit their capacity of self-regulation or executive functioning would be susceptible to or overwhelmed by the exposure of stressors. As such, to self-medicate psychological distress, individuals would resort to the drugs that offer a promptly change in their emotional state or positive emotion experiences (e.g., euphoria), and even develop dependence on the drugs. It is worth noting that COVID-19 long haulers can be significantly vulnerable to this maladaptive coping pathway since they encounter various challenges related to the symptoms, such as sick-leave, worry about health of themselves, career/job uncertainty, and social stigma [62–64]. Notably, the interruptions in mental health care due to the societal lock-down during the pandemic

would further aggravate their vulnerability [65]. These highlight the necessity to attend to psychiatric symptoms in developing care and treatment plans for COVID-19 long haulers.

Importantly, our results indicate that personal mastery and social support are negatively associated with substance use behaviors among COVID-19 long haulers, suggesting the protection of these psychosocial resources. This is consistent with previous substance use research, indicating that personal mastery was a critical cognitive resource to regulate distress, which was beneficial to abstinence or substance use reduction [32, 33, 35] (Majer, Jason, & Olson, 2004; Spencer & Patrick, 2009; Stoddard, Peirce, Hurd et al., 2020; Greenwood & Manning, 2016). Our finding is also aligned with a recent COVID-19 study, suggesting that social support was a protective factor against problematic substance use to cope with the pandemic [42]. The beneficial effects of these psychosocial resources appear to support the transactional theory of stress and coping [66]. This theory suggests that individuals' behavioral responses to stressors are dependent on their cognitive appraisal on the availability of resources to cope, while the resources can be intra- or interpersonal. If individuals perceived sufficient resources (e.g., high personal mastery and social support), they would experience lower negative moods and be unlikely to exhibit maladaptive coping (e.g., substance use). Given their protection from the risk, these two psychosocial resources have been identified as important factors in resilience-building intervention programs for people with chronic conditions [67–69]. Accordingly, to reduce substance use risk, resilience-based interventions tailored for distress management and psychosocial well-being enhancement would be warranted for COVID-19 long haulers.

There are several approaches that may be feasible to facilitate resilience factors and reduce substance use risk for COVID-19 long haulers. First, personal mastery and mental health outcomes could be improved through the implementation of mindfulness interventions with a focus no exploring intrinsic and extrinsic loci of control. Existing evidence has shown that the mindfulness-based trial is efficacious in controlling symptoms (e.g., pain severity), promoting psychological health, and reducing the risk for non-medical use of prescription opioids among patients with chronic pain [70]. Second, social support networks among COVID-19 long haulers may be developed through online social networking platforms, which encourage the fostering of communities among individuals who share the common experience of persisting COVID-19 symptoms. Indeed, our recent preliminary study of an online mindfulness-based walking intervention among COVID-19 long haulers suggested that the social-media- and mindfulness-based approaches were efficacious in pain control improvement and psychological health promotion (i.e., emotion regulation and stress relief) [71]. Future research would benefit from adopting these approaches to a large-scale trial among COVID-19 long haulers, in efforts to decrease the burden of psychiatric symptoms and substance use risk.

The current study had several methodological limitations. First, our model could not draw conclusions about causality due to the cross-sectional design. Second, our findings were based on self-reported data, which may be subject to response bias (e.g., social desirability). Third, some compounding effects were not controlled in the model since the relevant factors were not measured, such as pre-pandemic symptomatology of psychiatric symptoms and SUDs. Fourth, given a convenience sample on social media, the current study may be subject to self-selection bias and our findings have limited generalizability. Also, the Internet-based survey methodology would facilitate candor when assessing risk behaviors [72], this may be limited by only reaching COVID-19 long haulers on social media. Fifth, the PTSD measure in the current study showed low internal consistency (Cronbach's alpha of .48), although this is consistent with previous research on this scale [73]. To address these limitations, future research should consider applying a randomized sampling approach, a longitudinal design, a combination with in-person recruitment approach (e.g., at clinics), the collection from non-self-

reported data (e.g., health records), and an extensive screening of the history of psychiatric health disorders and SUDs.

Despite these limitations, as one of the first attempt to examine the substance use profile and relevant psychosocial factors among COVID-19 long haulers, the current study adds to the growing literature on psycho-behavioral influences of the COVID-19 pandemic. Our findings offer an extension to recent COVID-19 long hauler research, suggesting high prevalence not only in legal drug use (e.g., alcohol) but also in illicit drug use (e.g., Cocaine) and NMUPD (e.g., prescription opioids). Our SEM model suggests that substance use behaviors are driven by psychiatric symptoms, which has been a significant concern in COVID-19 long haulers. Importantly, personal mastery and social support appear to provide protection from the psychiatric influences on substance use, which may inform future research to reduce substance use risk among COVID-19 long haulers using approaches beneficial to these factors, such as mindfulness-based intervention and social-media-based trials.

## Author Contributions

**Conceptualization:** Cheuk Chi Tam, Shan Qiao, Alain H. Litwin, Xiaoming Li.

**Data curation:** Camryn Garrett.

**Formal analysis:** Cheuk Chi Tam, Camryn Garrett.

**Funding acquisition:** Xiaoming Li.

**Methodology:** Cheuk Chi Tam.

**Writing – original draft:** Cheuk Chi Tam, Shan Qiao, Camryn Garrett, Ran Zhang, Atefeh Aghaei, Abhishek Aggarwal.

**Writing – review & editing:** Cheuk Chi Tam, Shan Qiao, Alain H. Litwin, Xiaoming Li.

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
