## [Decision Letter · Decision Letter 0]

7 Mar 2023

PONE-D-22-32968Substance use, psychiatric symptoms, personal mastery, and social support among COVID-19 long haulers: A compensatory modePLOS ONE

Dear Dr. Tam,

Thank you for submitting your manuscript to PLOS ONE. After careful consideration, we feel that it has merit but does not fully meet PLOS ONE’s publication criteria as it currently stands. Therefore, we invite you to submit a revised version of the manuscript that addresses the points raised during the review process.

Please submit your revised manuscript by April 30, 2023.If you will need more time than this to complete your revisions, please reply to this message or contact the journal office at plosone@plos.org. Please include the following items when submitting your revised manuscript:A rebuttal letter that responds to each point raised by the academic editor and reviewer(s). You should upload this letter as a separate file labeled 'Response to Reviewers'.A marked-up copy of your manuscript that highlights changes made to the original version. You should upload this as a separate file labeled 'Revised Manuscript with Track Changes'.An unmarked version of your revised paper without tracked changes. You should upload this as a separate file labeled 'Manuscript'.

We look forward to receiving your revised manuscript.

Kind regards,

Colleen Anne Dell, Ph.D.

Academic Editor

PLOS ONE

Journal Requirements:

2. In your Methods section, please include additional information about your dataset and ensure that you have included a statement specifying whether the collection and analysis method complied with the terms and conditions for the source of the data.

“The research was supported in part by the by the National Institute of Allergy and Infectious Diseases of the National Institutes of Health under award number R01AI127203-5S1. The content is solely the responsibility of the authors and does not necessarily represent the official views of the National Institutes of Health.”

Additional Editor Comments (if provided):

Dear Cheuk Chi Tam and co-authors,

Thank you very much for your submission to PLOS ONE, short titled "Substance use and psychiatric and psychosocial factors among COVID-19 long haulers". I am sorry for the delay in the review - it was difficult to locate reviewers. Your manuscript has been accepted with minor revisions. They are listed below.

REVIEWER 1

• The paper is well-written and engaging. It is appropriately sourced and referenced.

• Authors have no bias.

• The study adequately addresses issues of potential gender bias.

• Strongly recommend that authors replace the term “substance misuse” with “substance use” throughout. The term “misuse” is now considered very pejorative (like the term “substance abuse”) and implies deliberate intentionality.

• Study strength – analytic strategy and techniques used. Very well formulated and executed.

• Study weakness – sampling technique – a sample of convenience using social media sources. This creates potential respondent bias compared to a randomly generated response sample.

REVIEWER 2

• Overall, the paper is about an interesting topic and the analysis is well described.

• In my opinion the paper deserves to be published in a better journal than Plos One. (for this the analysis should be extended and the presentation of the model improved)

• In general, write all numbers as 0.38 and not as .038!

• ...there have been 87,899,721 total cases of COVID-19 (4). Is this unique cases? Please clarify.

• The figures of the SEMs are very helpful and informative. If the figures could be in color that would be great!

• Results section 3.4 is too short and more explanations should be given about the analysis.

• It would be interesting to study the effect of the size of the data. To do this, subsampling could be used to study the effect on the results. What is the minimal sample size that allows to obtain the same results?

PLEASE NOTE: Because I do not have a statistical background, once your revisions are completed, it will be reviewed by a statistical expert at PLOS ONE.

Please let me know if you have any questions.

Sincerely,

Colleen Dell

Reviewers' comments:

Reviewer's Responses to Questions

**Comments to the Author**

1. Is the manuscript technically sound, and do the data support the conclusions?

Reviewer #1: Partly

Reviewer #2: Yes

2. Has the statistical analysis been performed appropriately and rigorously? 

Reviewer #1: Yes

Reviewer #2: Yes

3. Have the authors made all data underlying the findings in their manuscript fully available?

Reviewer #1: Yes

Reviewer #2: No

4. Is the manuscript presented in an intelligible fashion and written in standard English?

Reviewer #1: Yes

Reviewer #2: Yes

5. Review Comments to the Author

Reviewer #1: • The paper is well-written and engaging. It is appropriately sourced and referenced.

• Authors have no bias.

• The study adequately addresses issues of potential gender bias.

• Strongly recommend that authors replace the term “substance misuse” with “substance use” throughout. The term “misuse” is now considered very pejorative (like the term “substance abuse”) and implies deliberate intentionality.

• Study strength – analytic strategy and techniques used. Very well formulated and executed.

• Study weakness – sampling technique – a sample of convenience using social media sources. This creates potential respondent bias compared to a randomly generated response sample.

Reviewer #2: Overall, the paper is about an interesting topic and the analysis is well described.

In my opinion the paper deserves to be published in a better journal than Plos One. (for this the analysis should be extended and the presentation of the model improved)

In general, write all numbers as 0.38 and not as .038!

...there have been 87,899,721 total cases of

COVID-19 (4). Is this unique cases? Please clarify.

The figures of the SEMs are very helpful and informative. If the figures could be in color that would be great!

Results section 3.4 is too short and more explanations should be given about the analysis.

It would be interesting to study the effect of the size of the data. To do this, subsampling could be used to study the effect on the results. What is the minimal sample size that allows to obtain the same results?

6. PLOS authors have the option to publish the peer review history of their article (what does this mean?). If published, this will include your full peer review and any attached files.

Reviewer #1: **Yes: **John Weekes, Ph.D.

Reviewer #2: No

---

## [Author Response · Author response to Decision Letter 0]

16 Mar 2023

The editor and reviewer provided many insightful suggestions that we believe have improved the manuscript. Below are our detailed responses to reviewer’s comments (in italics).

Editor:

Responses: Thanks for the instructions. We have proofread the manuscript in line with PLOS ONE’s style requirements.

2. In your Methods section, please include additional information about your dataset and ensure that you have included a statement specifying whether the collection and analysis method complied with the terms and conditions for the source of the data.

Responses: Thanks for the suggestions. Data in the current study were collected by us and the collection and analysis process complied with the study protocol approved by the University of South Carolina Institutional Review Board, including obtaining online written consent prior to the participation and deidentification of the data. We have now noted in the manuscript on page 5. 

Responses: Thanks for catching that. We have now noted that online written consent was obtained from the participants prior to the surveys (on page 5). No minors were included in the study given the inclusion criteria of 18 years of age or older. As requested, we have added the text to the submission form.

“The research was supported in part by the by the National Institute of Allergy and Infectious Diseases of the National Institutes of Health under award number R01AI127203-5S1. The content is solely the responsibility of the authors and does not necessarily represent the official views of the National Institutes of Health.”

Responses: Given no additional funding support for this study, we have clarified that in the statement as requested (see below). The amended statement is also provided in the cover letter. 

‘The research was supported in part by the by the National Institute of Allergy and Infectious Diseases of the National Institutes of Health under award number R01AI127203-5S1. The content is solely the responsibility of the authors and does not necessarily represent the official views of the National Institutes of Health. There was no additional external funding received for this study.’

Responses: Thanks for the instructions. As requested, a dataset including variables of the current study has now been available on a repository. Below is the link to the data:

https://figshare.com/articles/dataset/SPSS_dataset_of_substance_use_in_COVID_long_haulers_sav/22285330

Responses: As requested, we have reviewed the in-text citations and reference thoroughly in the manuscript. 

Thank you very much for your submission to PLOS ONE, short titled "Substance use and psychiatric and psychosocial factors among COVID-19 long haulers". I am sorry for the delay in the review - it was difficult to locate reviewers. Your manuscript has been accepted with minor revisions. They are listed below.

Responses: We appreciate the editors and reviewers’ positive feedback on our manuscript. Also thanks for the valuable comments and suggestions, which have improved the manuscript. Please see our detailed responses to reviewers’ comments.

REVIEWER 1

• The paper is well-written and engaging. It is appropriately sourced and referenced.

• Authors have no bias.

• The study adequately addresses issues of potential gender bias.

Responses: Thanks for the positive comments. 

• Strongly recommend that authors replace the term “substance misuse” with “substance use” throughout. The term “misuse” is now considered very pejorative (like the term “substance abuse”) and implies deliberate intentionality.

Responses. Thanks for the suggestion. We have now used the term “substance use” throughout the manuscript. 

• Study strength – analytic strategy and techniques used. Very well formulated and executed.

Responses. Thanks for noting this.

• Study weakness – sampling technique – a sample of convenience using social media sources. This creates potential respondent bias compared to a randomly generated response sample.

Responses. We agree with the reviewer that the current study would be limited by the sampling approach (convenient sampling on social media). We have discussed the limitations and suggested that future research should use a randomized sampling approach and integrated forms of data (e.g., self-reported and electronic health records) to investigate substance use and its protective and risk factors in COVID-19 long haulers (on page 12).

REVIEWER 2

• Overall, the paper is about an interesting topic and the analysis is well described.

• In my opinion the paper deserves to be published in a better journal than Plos One. (for this the analysis should be extended and the presentation of the model improved)

Responses: Thanks for the comments.

• In general, write all numbers as 0.38 and not as .038!

Responses: We have made revisions as requested.

• ...there have been 87,899,721 total cases of COVID-19 (4). Is this unique cases? Please clarify.

Responses: Thanks for the question. This is the number of cumulative COVID-19 infection cases worldwide as of July 6, 2022. We have now clarified it in the manuscript on page 2.

• The figures of the SEMs are very helpful and informative. If the figures could be in color that would be great!

Responses: Thanks for the suggestion. We have colored the figures for better presentations.

• Results section 3.4 is too short and more explanations should be given about the analysis.

Responses: Thanks for the comment. In order to report concise and objective information in the results section, we left the explanation and interpretation of the SEM results in the discussion section. However, we briefly elaborated them a bit by noting the role of risk factors and protective factors based on the coefficients (on page 10).

• It would be interesting to study the effect of the size of the data. To do this, subsampling could be used to study the effect on the results. What is the minimal sample size that allows to obtain the same results?

Responses: Thanks for the suggestion. We have considered this suggested test, but we think it seems to be beyond the scope of our study and hesitate to do the analysis for the following consideration. Given the uncertainty of the representativeness and homoscedasticity across subsamples, the suggested analyses, by being run multiple times, would increase the threat to the Type I or Type II errors, affecting the robustness of the findings. Accordingly, the suggested analyses might not provide meaningful improvements to the study.

However, according to Kline (2010), the minimum sample size for SEM can be determined by the number of manifest factors, suggesting a minimum of 10 cases per one factor. In our structural model, we have a total of 18 manifest factors, implying a need of a minimum sample size of 180 to reach enough statistical power. The sample size in the current study was 460, which exceeded the requirement. 

Ref:

Kline, R. (2010). Principles and practice of structural equation modeling. Structural Equation Modeling. https://doi.org/10.1038/156278a0

We appreciate the editor’s and reviewers’ insightful comments and believe these revisions have strengthened the manuscript.

---

## [Decision Letter · Decision Letter 1]

19 Jul 2023

Substance use, psychiatric symptoms, personal mastery, and social support among COVID-19 long haulers: A compensatory mode

PONE-D-22-32968R1

We’re pleased to inform you that your manuscript has been judged scientifically suitable for publication and will be formally accepted for publication once it meets all outstanding technical requirements.

Kind regards,

Colleen Anne Dell, Ph.D.

Academic Editor

PLOS ONE

Additional Editor Comments (optional):

Reviewers' comments:

Reviewer's Responses to Questions

**Comments to the Author**

1. If the authors have adequately addressed your comments raised in a previous round of review and you feel that this manuscript is now acceptable for publication, you may indicate that here to bypass the “Comments to the Author” section, enter your conflict of interest statement in the “Confidential to Editor” section, and submit your "Accept" recommendation.

Reviewer #1: All comments have been addressed

Reviewer #3: All comments have been addressed

2. Is the manuscript technically sound, and do the data support the conclusions?

Reviewer #1: Yes

Reviewer #3: (No Response)

3. Has the statistical analysis been performed appropriately and rigorously? 

Reviewer #1: Yes

Reviewer #3: (No Response)

4. Have the authors made all data underlying the findings in their manuscript fully available?

Reviewer #1: Yes

Reviewer #3: (No Response)

5. Is the manuscript presented in an intelligible fashion and written in standard English?

Reviewer #1: Yes

Reviewer #3: (No Response)

6. Review Comments to the Author

Reviewer #1: The authors have adequately addressed all concerns raised. The manuscript is now ready to move forward.

Recommend accept.

Reviewer #3: (No Response)

7. PLOS authors have the option to publish the peer review history of their article (what does this mean?). If published, this will include your full peer review and any attached files.

Reviewer #1: **Yes: **John Weekes, Ph.D.

Reviewer #3: No

---

## [Editor Report · Acceptance letter]

26 Jul 2023

PONE-D-22-32968R1 

Substance use, psychiatric symptoms, personal mastery, and social support among COVID-19 long haulers: A compensatory model 

Dear Dr. Tam:

I'm pleased to inform you that your manuscript has been deemed suitable for publication in PLOS ONE. Congratulations! Your manuscript is now with our production department. 

Kind regards, 

on behalf of

Dr. Colleen Anne Dell 

Academic Editor

PLOS ONE